# Disentangling covariates to predict counterfactuals for single-cell data

## Abstract

Single-cell transcriptomics enables understanding cellular behaviors during diseases or in response to perturbations. However, analyzing multi-donor and multi-covariate single-cell data while disentangling technical noise from biological signals remains a significant challenge. Additionally, predicting cellular responses to interventions becomes even more challenging due to donor-specific effects and unobserved covariates. This study introduces *Disentangling covariates to predict counterfactuals (dis2p)*, a causal generative model designed to disentangle known covariate variations from unknown ones while simultaneously learning to make counterfactual predictions. dis2p accurately learns covariate-specific representations, as empirically demonstrated, which improve generalization for performing counterfactual predictions. Given the increasing availability of population-level single-cell datasets, we envision dis2p becoming a valuable tool for analyzing such data due to its ability to learn controllable representations that facilitate biological discoveries, improve experiment design, and reduce costs using in silico predictions. [1]

## 1 Introduction and related work

Single-cell technologies enable profiling gene expression at the resolution of individual cells providing a high-resolution understanding of disease Sikkema et al. (2023a); Ahern et al. (2022) and human development Herring et al. (2022); Haniffa et al. (2021) and response to perturbations Dixit et al. (2016); Replogle et al. (2020); Schraivogel et al. (2020).

The community has collaborated to create consortia such as the Human Cell Atlas (Regev et al. (2018)), aiming to construct a comprehensive map of cells throughout the human body by developing integrated cell atlases for individual organs. These cell atlases encompass millions of cells (Litviňuková et al. (2020); Kanemaru et al. (2023); Sikkema et al. (2023b); Salcher et al. (2022)) from diverse donors, laboratories, and experimental protocols, resulting in sample-specific attributes often known as batch effects. The primary objective in single-cell biology is to eliminate these unwanted effects while preserving the biological signal, as emphasized in Luecken et al. (2022). This task is further complicated by gene expression being a high-dimensional vector, the relatively limited number of samples, and the existence of numerous unobserved covariates, constituting an individual-specific background.

As a simple example, consider scRNA datasets with two covariates: disease state (healthy or diseased) and batch ID. To study the disease, our aim is to identify novel (pathological) cell types unique to diseased individuals. One approach is to create a latent space that specifically captures disease effects. In this space, the new population of cells associated with the disease should form a distinct cluster. To validate that the disease latent space is indeed free from batch effects, we propose creating a separate latent space solely for batch effects as a negative control. In this control space, the new population of cells should not form a distinct cluster. If they do, it would indicate that the clustering of these cells together is driven by batch effects Lotfollahi et al. (2022); Dann et al. (2022).

---

[1]The code for the method is available at https://shorturl.at/ptJR6 and for the analysis is at https://shorturl.at/ejGV9

Existing single-cell methods for data integration/batch removal either infer a single latent space where batch effects are regressed out, as shown in Lopez et al. (2018); Xu et al. (2021); Donno et al. (2022); Korsunsky et al. (2019); Welch et al. (2019), or infer two or more latent spaces, as demonstrated in Weinberger et al. (2023); Zhang et al. (2023b). For instance, ContrastiveVI, a variational auto-encoder Kingma et al. (2019), can model one covariate at a time, learning a shared latent space where the common variability of that covariate and others is preserved, and a distinct one where only the variability of the target covariate is preserved.

Furthermore, scDisInFact Zhang et al. (2023b) learns a shared latent space where batch ID is regressed, and multiple unshared ones where covariate variations are preserved while batch ID is removed. However, none of these models provide a negative control latent space where batch effects are preserved, and other covariates are regressed out. This is essential to evaluate if the effects observed in the covariate space are due to batch effects or biological variability. Many of these methods draw inspiration from general approaches for supervised fair and disentangled representation learning, such as Flexibly Fair Representation Learning by Disentanglement (FFVAE) Creager et al. (2019) and Fader networks Lample et al. (2017), or unsupervised disentanglement learning approaches like $\beta$-VAE Higgins et al. (2016) and $\beta$-TCVAE. While we compare and benchmark some of these approaches, our focus here is to compare against existing methods specifically designed to model single-cell data.

Despite advancements in experimental technology, screening many interventions such as drugs and genetic perturbations, or studying diseases across donors with varying backgrounds at the single-cell level, remains a challenge due to the associated experimental costs Srivatsan et al. (2020); Lotfollahi et al. (2021). This challenge has led to the development of methods Ji et al. (2021) aimed at predicting out-of-distribution (OOD) data for covariates not measured in the initial experiment. This approach helps narrow down the hypothesis space and facilitates efficient biological discoveries. For example, GEARS Roohani et al. (2023) can predict genetic perturbations, but it lacks any theoretical foundation and cannot handle multiple covariates, including batch effects, potentially leading to spurious predictions.

On the other hand, the compositional perturbation auto-encoder Lotfollahi et al. (2021) and its extension ChemCPA Hetzel et al. (2022) can handle drugs and other types of perturbations. However, they assume strong linearity in the latent space, limiting their ability to capture non-linear effects. While CPA, in practice, can perform counterfactual-like predictions, there is no explicit element or theory explaining why the model should provide accurate counterfactual predictions. Moreover, Variational Causal Inference Wu et al. (2022) models this challenge as using variational inference but can handle only one covariate as the perturbation at a time. It cannot generalize to counterfactual predictions for multiple covariates. In a similar vain, Foster et al. (2022) combines fairness and counterfactual inference that is based on a new way of imposing disentanglement with theoretical guarantees. However the disentanglement theory they develop comes at a cost of studying a relatively simple SCM with one covariate which again implies that they cannot flexibly change the covariate at test time. Furthermore, cellOT Bunne et al. (2021) uses optimal transport to map control cells to perturbed cells but is limited to model one covariate and it is not a generative model. Finally, Biolord leverages a unique cell embedding for supervised disentanglementPiran et al. (2023). It uses a compositional approach similar to CPA, altering composition during testing to predict unseen covariates. However, it lacks loss terms and theory to encourage counterfactual and OOD prediction. Moreover, it struggles to scale to large datasets due to exploding parameters caused by per-cell optimization instead of amortized regimes.

Finally, causal disentanglement aims to learn the graph relating the causal variables. Several identifiability theories have led to methods in single-cell data Seigal et al. (2022); Zhang et al. (2023a). However, this comes at the expense of introducing linearity assumptions about the structural equations in the models Lopez et al. (2023) or very technical and unverifiable assumptions about the graph and the probability distributions Lachapelle et al. (2022); Lopez et al. (2023); Lachapelle & Lacoste-Julien (2022). All of these assumptions can reduce the accuracy of out-of-distribution predictions if they are not satisfied.

To address these challenges, we propose a new model that extends previous work on variational causal inferenceWu et al. (2023) to predict counterfactuals for multiple covariates simultaneously. This model also provides a disentangled representation necessary to distinguish technical and biological signals in multi-covariate single-cell datasets. Our main contributions are following:

- We introduce *dis2p*, a variational causal model that facilitates comparative analyses of single-cell data by disentangling the effects of biological and technical covariates.

- We demonstrate that dis2p can predict single-cell gene expression for odd values of any of those covariates by predicting counterfactual outcomes for multiple covariates, which is not possible with existing causal prediction models.

- We demonstrate that dis2p outperforms existing methods in both disentangled representation learning for both uni-covariate and multi-covariate scenarios, as well as OOD counterfactual predictions.

## 2 DISENTANGLING COVARIATES TO PREDICT COUNTERFACTUALS (DIS2P)

We consider a dataset $\mathcal{D} = \{(X^{(j)}, S^{(j)}\}_{j=1}^N$, where each $X^{(j)} \in \mathbb{R}^G$ describes the gene expression of $G$ genes from cell $j$. We assume the data has $n$ observed covariates. The covariate vector $S^{(j)} = (S_i^{(j)}, \ldots, S_n^{(j)})$ contains elements $S_i^{(j)}$ describing the value of covariate $i$ in cell $j$. These can include covariates of a categorical or continuous type such as cell type, age, states of diseases, treatments, gene embeddings, etc. and each can take multiple values. Throughout the rest of the paper when we refer to a specific cell or generally the random variable, we will drop the superscript $(j)$.

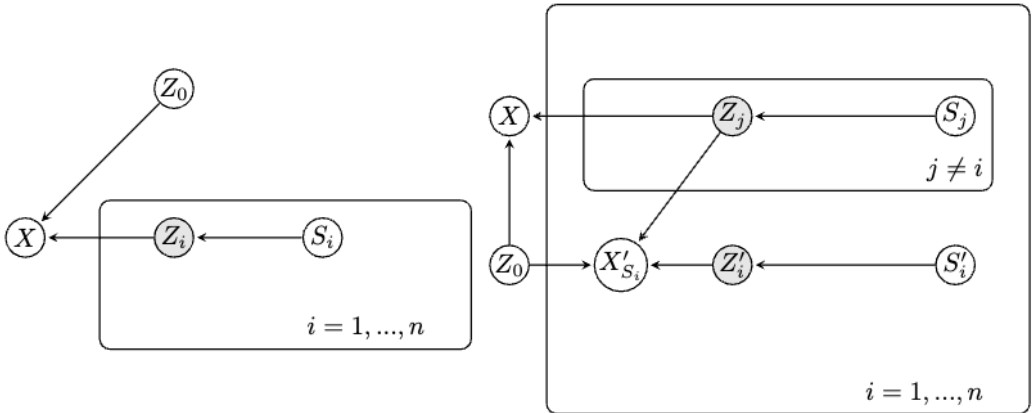

Figure 1: The high-level architecture and counterfactual prediction procedure for dis2p. (a) The cell encoders receive gene expression ($X_G$) for each cell along with its covariates to infer covariate-specific latent variables ($z^i$), which are concatenated with covariate vectors to reconstruct the gene expression again ($\hat{X}_G$). (b) The latent variable $z^i$ is fed into a classifier to discriminate its values, while the concatenation of all other latent variables except $z^i$ itself, denoted as $z^{-i}$, is fed into an adversarial classifier for covariate $i$. (c) Counterfactual prediction of gene expression $x''_G$ from double intervention $S_1, S_i \rightarrow S'_1, S'_i$ based on abduction from the parallel world diagrams (see Figure A).

We assume that our dataset $\mathcal{D}$ was produced from the Structural Causal Model (SCM) of Figure 1. A SCM is an ordered triple $\langle U, V, E \rangle$, where U is a set of exogenous variables whose values are determined by factors outside the model; V is a set of endogenous variables whose values are determined by factors within the model; and $E$ is a set of structural equations that express the value of each endogenous variable as a function of the values of the other variables in $U$ and $V$.

In our case $U = \{Z_0\}$, $V = \{S_i, Z_i : i = 1, \ldots, n\}$ and we don't impose any assumption about the functional form of the structural equations $E$, in contrast to several causal disentanglement approaches Seigal et al. (2022); Lopez et al. (2023); Lachapelle et al. (2022).

In other words, each covariate $S_i$ causes a latent variable $Z_i$ that captures all the aspects of that covariate. To give an example, age is just a number but it can manifest in different ways in the skin, brain, heart, etc. all of which are summarized into a latent age variable $Z_{age}$. In addition, there

is a background latent variable $Z_0$ that captures information not contained in the covariates in our dataset. Then all of these latent variables cause the response variable $X$.

In this work we are also interested in reliably predicting counterfactuals, which are the hypothetical values $X_{s_i}^{(j)}$ the gene expression would have assumed for cell $j$ had the value of its $i$-th covariate been $S_i^{(j)} = s_i$. In other words, we are after individual-level effects which as argued in [cite Causality Pearl] further motivates the need for the latent variable $Z_0$ that summarizes the identity and individuality of the cell. Using this Structural Causal Model and Pearl's twin (or many-world) graphs Pearl (2009) such as the one in Figure 1 we can predict counterfactuals and correlations between counterfactual and factual variables using the d-separation calculus, which we then use to train our model.

For categorical covariates, we use a one-hot encoding such that if $S_k$ accepts $d_k$ different values then it is encoded into a $d_k$-dimensional one-hot vector.

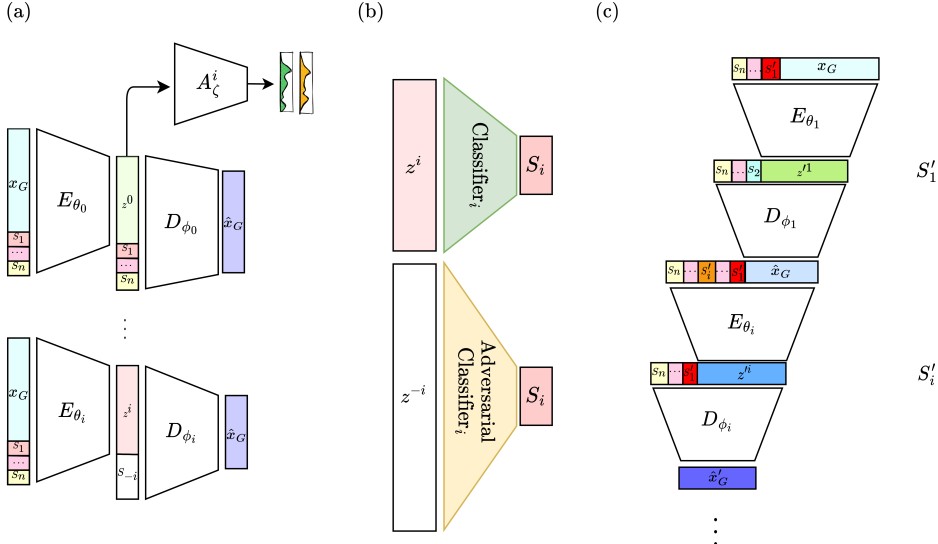

Figure 2: The high-level architecture and counterfactual prediction procedure for dis2p. (a) The cell encoders receive gene expression ($X_G$) for each cell along with its covariates to infer covariate-specific latent variables ($z^i$), which are concatenated with covariate vectors to reconstruct the gene expression again ($\hat{X}_G$). (b) The latent variable $z^i$ is fed into a classifier to discriminate its values, while the concatenation of all other latent variables except $z^i$ itself, denoted as $z^{-i}$, is fed into an adversarial classifier for covariate $i$. (c) Counterfactual prediction of gene expression $x_G''$ from double intervention $S_1, S_i \rightarrow S_1', S_i'$ based on abduction from the parallel world diagrams (see Figure 1).

dis2p architecture and model has four main components : (1) cell encoders, (2) cell decoders, (3) the covariate normal and adversarial classifiers, and (4) the counterfactual prediction procedure, as illustrated in Figure 2.

## 2.1 CELL ENCODERS

We use $n + 1$ encoders parameterized by $\phi_0, \phi_1, \ldots, \phi_n$ to separately infer $n + 1$ latent vectors $Z_0, Z_1, \ldots, Z_n$. All encoders share the same input: the gene expression vector $X$ and the full covariate vector $S$. We also have $n$ encoders parameterized by $\psi_1, \ldots, \psi_n$ to separately infer $n$ prior distributions for latent variables. The input of the encoder $\psi_k$ is only $S_k$.

We use amortized inference Kingma et al. (2019); Lopez et al. (2018) of the latent variable $Z_k$ as follows:

$$q_{\phi_k}(Z_k \mid X, S) \sim \text{Normal}\left(g_\mu(X, S), \text{diag}(g_\sigma^2(X, S))\right).$$

In accordance to Proposition 1 (proof in the Appendix Note X), we also infer a prior distribution for $Z_k$ as follows:

$$q_{\psi_k}(Z_k \mid S_k) \sim \text{Normal}\left(g_\mu(S_k), \text{diag}(g_\sigma^2(S_k))\right).$$

## 2.2 Cell decoders

We use $n+1$ decoders parameterized by $\theta_0, \theta_1, \ldots, \theta_n$ to generate the gene expression vector $X$ separately in each decoder. The input of decoder 0 is $(Z_0; S)$, and for the $k$-th decoder (where $k \neq 0$) is $(Z_k, S_{-k})$, where $S_{-k}$ denotes all attributes except $S_k$.

This is motivated by our aim to make $Z_k$ capture all the information in $S_k$ and none of the information in $S_{-k}$, where $k \neq 0$ (see next section). In this way, the very architecture of dist2p is conducive to learning disentangled representations. Similarly, we want $Z_0$ not to capture any information about $S$, and instead learn cell-specific background variation (due to unobserved covariates).

Using decoder $\theta_k$ we predict the gene expression of gene g $X_g$ with a Zero Inflated Negative Binomial (ZINB) (or alternatively a Negative Binomial (NB)) which has been shown to correctly model gene expression Lopez et al. (2018): distribution can be used parameterized by $\theta_k$. The gene expression for gene $g$ in cell $j$ is given by:

$$X_g \sim \text{NegativeBinomial}\left(l f^g(Z), d_g\right),$$

where $l$ is the number of captured number of RNA molecules captured for that gene in the cell, which we assume to be an observed random variable. $f$ is a function mapping the latent space to the simplex of the gene expression space. $d_g$ is the dispersion parameter of the negative binomial distribution. ZINB has one additional parameter the mixture coefficient that represents the weight of the point mass. All these parameters are learned using linear neural networks. By default, we use ZINB in the generative process because it is more suitable for count data that contains a vast majority of zeros as in gene expression count data Lopez et al. (2018).

## 2.3 Covariate classifiers

We seek the following properties for the latent space:

1. $MI(Z_i, S_i)$ is high $\forall i$ (predictability of $S_i$ given $Z_i$).
2. $MI(Z_{-i}, S_i)$ is low $\forall i$ (unpredictability of $S_i$ given $Z_{-i}$).

where $MI$ stand for mutual information defined as $MI(X, Y) = \sum_{x \in X} \sum_{y \in Y} P(x, y) \log\left(\frac{P(x,y)}{P(x)P(y)}\right)$. To achieve property (1), for each $i$, we will use a classifier that predicts $S_i$ given $Z_i$ and penalize our loss function with a cross-entropy loss $CE(\text{classifier}(S_i|Z_i), S_i)$ of these classifiers. To achieve property (2), for each $i$, we will use a classifier that predicts $S_i$ given $Z_{-i}$, and penalize our loss function with cross-entropy loss $CE(\text{classifier}(S_i|Z_{-i}), S_i)$ of these classifiers.

All classifiers are fully connected networks with one hidden layer of size 128 by default, and predict the one-encoding for the given attribute.

## 2.4 Counterfactual inference

Given a cell with expression and covariate values $(X, S)$, the objective is to predict the counterfactual $X'$ when new attributes $S' = S_{-k} \cup S'_k$ are provided. This is achieved by estimating the counterfactual distribution using the equation:

$$\hat{p}_k(X'|X, S', S) := \int q_{\phi_k}(Z|X, S') p_{\theta_k}(X'|Z, S_{-k}) dZ$$

Here, the encoder $k$ is employed to infer a latent vector $Z'_k$ by taking inputs $(X, S')$. This latent vector $Z'_k$ is randomly sampled from the distribution $q_{\phi_k}(Z|X, S')$. Subsequently, $Z'_k$ is passed to decoder $k$ along with other attributes $S_{-k}$ to generate the distribution $p_{\theta_k}(X'|Z, S_{-k})$.

Assume we are given a data sample $(x, s)$, and we want to predict the counterfactual $x'$ if new attributes $s' = \bigcup_{k=1}^{n} s'_k$ were given. In this case, we choose a random permutation for $\{1, 2, \ldots, n\}$ like $\{\pi(1), \pi(2), \ldots, \pi(n)\}$. Then we use the specified ordering of our $n$ covariate-specific networks (see Figure 2) by this random permutation (with latent $Z_1, \ldots, Z_n$). The input for network $k$ is $X_{k-1}$ along with attributes $\tilde{S}_k = \{S'_{\pi(1)}, \ldots, S'_{\pi(k)}\} \bigcup \{S_{\pi(k+1)}, \ldots, S_{\pi(n)}\}$. We take the mean of the ZINB distribution $\hat{p}_k(X_k | Z_k, \tilde{S}_k)$ generated by network $k$ as $\tilde{X}_k$ and give it as input to network $k + 1$ along with attributes $\tilde{S_{k+1}} = \{S'_{\pi(1)}, \ldots, S'_{\pi(k+1)}\} \bigcup \{S_{\pi(k+2)}, \ldots, S_{\pi(n)}\}$. Finally, $x'$ is predicted as the mean of the ZINB distribution $\hat{p}_n(X_n | Z_n, \tilde{S}_n)$.

Note that in the given counterfactual attributes, there might exist some indices $k$ with $s'_k = s_k$. If we know these indices, we can remove them from the above process to optimize the number of steps and reduce error propagation. However, in the training process, we pair random samples randomly to avoid time-consuming processing of the attributes of samples, and we have no information about possible equal attribute values of pairs. Therefore, we apply the aforementioned process to predict counterfactual gene expression.

## 2.5 Loss function

As explained in Appendix A (see Corollary 1), based on the structure of our assumed Structural Causal Model we derive a lower bound on the sum of the (factual) posterior likelihood and counterfactual posterior likelihood, which translates into the following loss function,

$$
\begin{aligned}
\mathcal{L}(\theta, \phi) = &- \sum_{i=0}^{n} \mathbb{E}_{q_{\phi_i}(Z_i|X,S)} \log p_{\theta_i}(X|Z_i, S_{-i}) - \alpha_1 \sum_{i=1}^{n} \log \hat{p}(X'|X, S', S) \\
&+ \sum_{i=1}^{n} D(q_{\phi_i}(Z_i|X,S) \parallel q_{\psi_i}(Z_i|S_i)) + \alpha_2 \sum_{i=1}^{n} CE(classifier(S_i|Z_i), S_i) \\
&- \alpha_3 \mathcal{L}_{adv}(\theta, \phi),
\end{aligned}
\tag{1}
$$

where $\hat{p}(X'|S')$ is the likelihood of the $X'$ in the counterfactual distribution derived from the method explained in Section 2.4, $\mathcal{L}_{adv}(\theta, \phi) = \sum_{i=1}^{n} CE(classifier(S_i|Z_{-i}), S_i)$, and $\alpha_1, \alpha_2, \alpha_3$ are heuristic hyperparameters which we explore in the ablation experiment.

# 3 Metrics

## 3.1 Disentanglement metrics

We divide disentanglement benchmarks into two scenarios. The first scenario has two covariates: a batch ID and a biological covariate. This is equivalent to a batch removal task in the field. The ideal method should remove the batch ID effect and preserve the biological covariate, which in this case is cell-type labels. We use standard metrics called scIB for each category as proposed by Luecken et al. (2022).

The second scenario extends the number of covariates beyond two, and only specific methods can handle such cases. We benchmark our method using the Mutual Information Gap [Chen et al. (2018); Higgins et al. (2016); Wu et al. (2023); Kumar et al. (2017); Kim & Mnih (2018)] due to its ability to generalize and be unbiased [Chen et al. (2018); Sepliarskaia et al. (2021)]. Specifically, we use two variations of MIG:

- $\text{maxMIG}(Z_1, ..., Z_n; S_1, .., S_n) = \frac{1}{n} \sum_{i=1}^{n} \frac{1}{H(S_i)} \max_{j \neq i} \left[ \text{MI}(Z_i, S_i) - \text{MI}(Z_i, S_j) \right]$

- $\text{catMIG}(Z_1, ..., Z_n; S_1, .., S_n) = \frac{1}{n} \sum_{i=1}^{n} \frac{1}{H(S_i)} \left[ \text{MI}(Z_i, S_i) - \text{MI}(Z_{-i}, S_i) \right]$

It's worth noting that, by its very definition, catMIG can assume negative values. For example, when all the non-age latent spaces together are more informative about age than the age latent space.

## 3.2 OOD METRICS

To evaluate the counterfactual gene expression prediction, we use the Pearson correlation $R^2$ between the ground truth values of held-out datasets and the predicted values, which is a widely used metric for this task Lotfollahi et al. (2021); Wu et al. (2022). We report $R^2$ for both all genes and the top 20 differentially expressed genes (DEGs).

## 4 EXPERIMENTS

The hyperparameters for all experiments are described in Appendix B.

### 4.1 DESCRIPTION OF DATASETS

To benchmark our model, we utilized three real scRNA-seq datasets, which we refer to as *Heart*, *Blood*, and *Liver*. The *Heart* atlas datasetLitviňuková et al. (2020) consists of approximately 485K cells from 14 donors, covering 11 major cell types. For benchmarking purposes, this dataset was subsampled to 18,641 cells. The covariates for each cell in this dataset include its type, source, the gender of its donor, and its location in the tissue. The *Blood* datasetKang et al. (2018) comprises 24,673 cells from control and interferon (IFN)-$\beta$ stimulated cells, which we treat as a binary covariate, and a categorical covariate indicating the cell type. Finally, the *Liver* dataAfriat et al. (2022) contains 19,053 infected and uninfected hepatocytes at various time points, along with their inferred spatial coordinates from mice.

### 4.2 DISENTANGLEMENT EXPERIMENT

To evaluate the disentanglement capabilities of dis2p, we conducted comprehensive assessments using the SCIB package and the Mutual Information Gap (MIG) metric. The SCIB package was employed to assess disentanglement between the primary latent space and the batch latent space. Meanwhile, the MIG metric, a widely recognized measure for evaluating multi-covariate disentanglement, was used for a holistic assessment.

| Method | Bio conservation | | | | | Batch correction | | | | | Aggregate score | | |
| | Isolated labels | KMeans NMI | KMeans ARI | Silhouette label | cLISI | Silhouette batch | iLISI | KBET | Graph connectivity | PCR comparison | Batch correction | Bio conservation | Total |
|---|---|---|---|---|---|---|---|---|---|---|---|---|---|
| dis2p | 0.62 | 0.90 | 0.91 | 0.72 | 1.00 | 0.97 | 0.86 | 0.72 | 0.96 | 1.00 | 0.90 | 0.83 | 0.86 |
| Harmony | 0.61 | 0.81 | 0.84 | 0.66 | 1.00 | 0.94 | 0.73 | 0.61 | 0.91 | 1.00 | 0.84 | 0.78 | 0.81 |
| PCA | 0.61 | 0.81 | 0.83 | 0.66 | 1.00 | 0.93 | 0.69 | 0.53 | 0.92 | 1.00 | 0.81 | 0.78 | 0.80 |
| LIGER | 0.61 | 0.61 | 0.41 | 0.62 | 1.00 | 0.93 | 0.82 | 0.73 | 0.88 | 0.99 | 0.87 | 0.65 | 0.76 |
| scANVI | 0.59 | 0.69 | 0.48 | 0.61 | 1.00 | 0.93 | 0.53 | 0.34 | 0.97 | 0.83 | 0.72 | 0.67 | 0.70 |
| scVI | 0.57 | 0.54 | 0.37 | 0.58 | 1.00 | 0.93 | 0.53 | 0.35 | 0.97 | 0.78 | 0.71 | 0.61 | 0.66 |

Figure 3: Disentanglement between the cell-type and the batch covariate latent spaces in the *Blood* dataset.

As illustrated in Figure 3, our approach demonstrated superior performance compared to alternative methods spanning neural network and statistical approaches used by the community across various evaluation metrics. This performance improvement was observed both in the overall disentanglement and in nearly all individual submetrics when tested on *Blood dataset*. It is worth noting that the primary goal in computational biology is the removal of batch effects while preserving the biological signal hence our algorithm will be of great value to the computational biology community.

Furthermore, our algorithm exhibited superior performance in terms of maxMIG and catMIG, as illustrated in Figure 4. Detailed definitions of these metrics can be found in the Metrics section. In Figure 4a, we can visually observe how our supervised learning approach for covariate-specific latent spaces enabled the model to effectively classify individual covariates while remaining agnostic to other covariates.

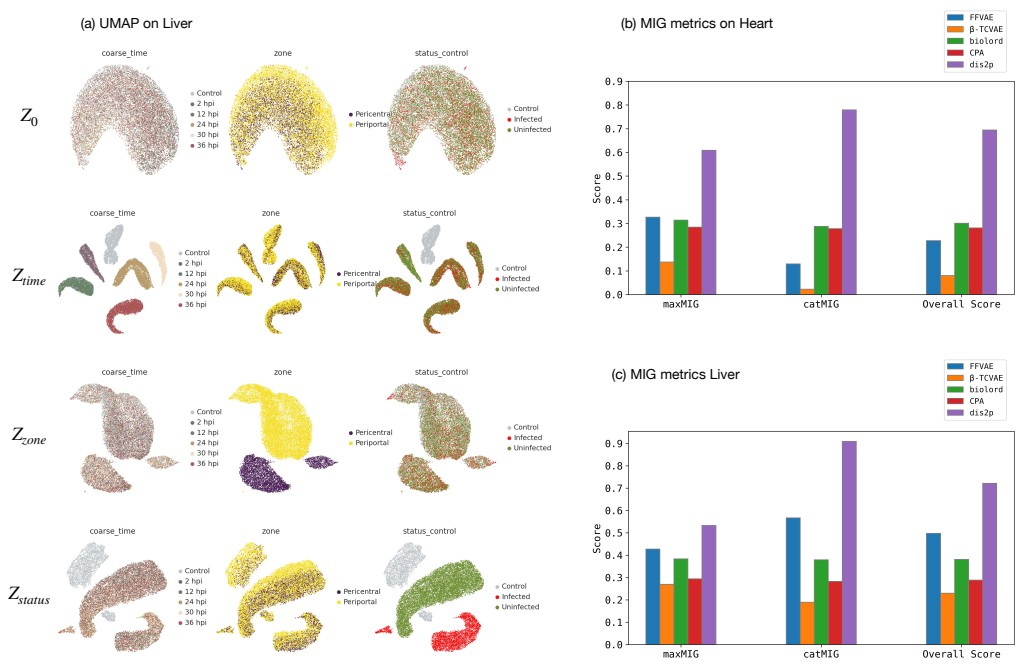

Figure 4: (a) UMAP visualization of disentanglement across all covariates in the *Liver* dataset. (b-c) Comprehensive disentanglement metrics on *Heart* and *Liver*. The overall score reported is the mean of maxMIG and catMIG.

### 4.3 COUNTERFACTUAL OOD PREDICTIONS

We assess the counterfactual prediction ability of dis2p using the 'Liver' and 'Heart' datasets. The 'Liver' dataset comprises three covariates: 'coarse_time' (6 unique values), 'zone' (2), and 'status_control' (3). Our interest lies in predicting disease progression over time. Hence, we partitioned the data, withholding cells where 'zone=Periportal' and 'status_control=Infected' for out-of-distribution (OOD) testing, and trained dis2p. During inference, we inputted all OOD cells with 'coarse_time=12 hours post infection (hpi)' to predict the appearance of infected cells ('status_control=Infected') at 'zone=Periportal' at the latest time post-infection ('coarse_time=36 hpi'). Our observations revealed that dis2p accurately predicted the disease response at this previously unseen time, compared to the ground truth of disease cells at that time point, validated using Differentially Expressed Genes (DEGs) ($R^2 = X$, Figure 5a).

To showcase the generalization of our method, we trained dis2p on the 'Heart' dataset, encompassing the attributes: 'cell_type' (11 unique values), 'cell_source' (4), 'gender' (2), and 'region' (6), where 'cell_source' and 'region' denote different labs and heart regions each cell belongs to. Our focus here is to model gender differences for a given cell type. Thus, we retained all cells across both genders with 'cell_type=Ventricular_Cardiomyocyte', 'cell_source=Sanger-Nuclei', and 'region=RV'. Our results demonstrate that dis2p accurately predicted the gender effect for cells from male donors, predicting as if they were female ($R^2 = X$, Figure 5b). We compared our method with two state-of-the-art (SOTA) methods, CPA and Biolord, which can handle multiple covariates using their default parameters recommended by the authors. Overall, we observed a significant improvement in our method's out-of-distribution prediction results, as depicted in Figure5c, attributing this enhancement to explicitly modeling the prediction problem in the context of causality and disentanglement.

## 5 GENERALIZATION AND DISENTANGLEMENT

We aimed to evaluate the relationships between the counterfactual term and classifier term encouraging disentanglement. To achieve this, we varied the coefficients from small to larger values using the

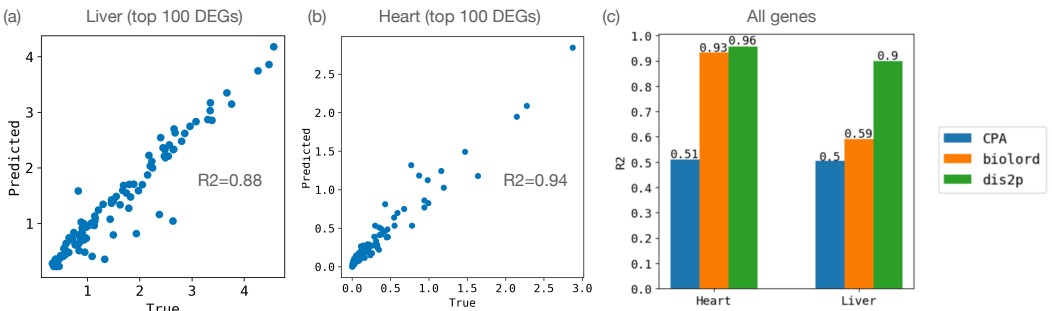

Figure 5: (a-b) ds2p predictions evaluated using the top 100 DEGs for Liver and Heart data. (c) Benchmarking against existing methods evaluated using all genes.

*Heart* dataset. The counterfactual splits were selected in a manner similar to previous results. During this exploration, we observed an interesting pattern: the best OOD predictions were achieved with higher disentanglement coefficients (Figure 6). This observation suggests that different loss components are necessary for accurate prediction, and it empirically demonstrates that better-disentangled representations lead to increased generalization. We hypothesize that accounting for other covariates can reduce noise and confounding effects when intervening on a specific covariate leading to such phenomena.

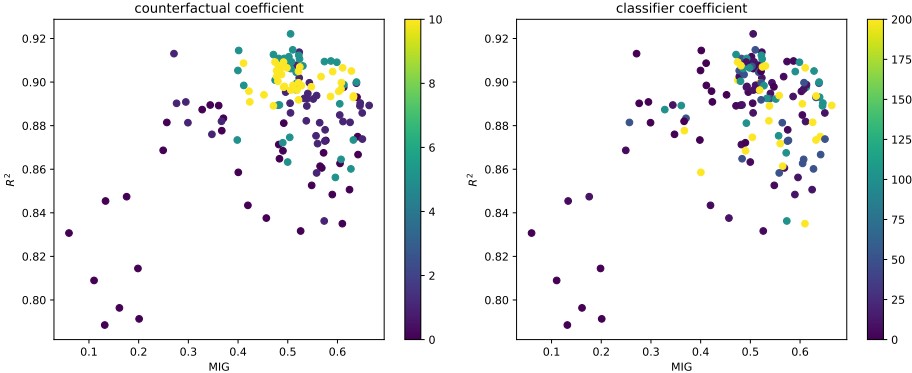

Figure 6: Relationship between loss components and generalization.

## 6    CONCLUSION

We introduced dis2p, a novel generative model designed to learn disentangled representations and make causal counterfactual predictions. We demonstrated the superiority of our model by comparing it with existing state-of-the-art models across a variety of tasks. Additionally, we conducted ablation studies that illustrate the relationship between disentanglement and out-of-distribution (ODD) generalizations. Furthermore, we expanded dis2p to handle multiple modalities, in addition to gene expression, such as chromatic accessibility. This extension enables the model to compare cells across covariates and latent spaces. Moreover, incorporating external gene embeddings Roohani et al. (2023) or drug embeddings, instead of one-hot vectors, allows for generalization to unseen perturbations Hetzel et al. (2022). We believe that dis2p will emerge as an essential tool for the single-cell community, facilitating interpretable analysis and prediction of single-cell data. This, in turn, will lead to novel biological discoveries and hypothesis generation.

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

## A  APPENDIX A: METHOD DERIVATION

Our modeling of the gene expression generating process assumes that each of the different attributes causes a latent variable which then causally affects the gene expression as illustrated in the structural causal model (SCM) in Figure 1. To consider the counterfactual case where a given cell had a different value for one of its attributes we use the many worlds graph in Figure 1, where we have also included a background variable U (aka $Z_0$) that summarizes any hidden attributes to which we don't have access but remain they same across worlds, thereby forming the "identity" of the cell. For counterfactual predictions involving changes to two attributes of a cell we need the many worlds graph of Figure 3 and similarly for more simulatneous changes of attributes.

**Proposition 1**
For any $i \in \{1, 2, \ldots, n\}$, given the causal structure in Figure 1, we have:

$$\log p(X|S) = \mathbb{E}_{p(Z_i|X,S)} \log p(X|Z_i, S) \\ - D(p(Z_i|X, S) \parallel p(Z_i|S_i)) \tag{2}$$

where $S = S_{-i} \cup S_i$.

**Proof**
According to the Bayes rule, we have:

$$p(Z_i|X, S)p(X|S) = p(X|Z_i, S)p(Z_i|S)$$

From the causal graph in Figure 1 or Figure 3, we know $Z_i \perp S_{-i}|S_i$, which implies $p(Z_i|S) = p(Z_i|S_i)$, therefore:

$$p(Z_i|X, S)p(X|S) = p(X|Z_i, S)p(Z_i|S_i)$$

By rewriting it and taking average over $p(Z_i|X, S)$, we have:

$$\mathbb{E}_{p(Z_i|X,S)} \log \frac{p(Z_i|X, S)}{p(Z_i|S_i)} = \mathbb{E}_{p(Z_i|X,S)} \log \frac{p(X|Z_i, S)}{p(X|S)}$$

$$D(p(Z_i|X, S) \parallel p(Z_i|S_i)) = \mathbb{E}_{p(Z_i|X,S)}[\log p(X|Z_i, S) - \log p(X|S)]$$

$$D(p(Z_i|X, S) \parallel p(Z_i|S_i)) = \mathbb{E}_{p(Z_i|X,S)} \log p(X|Z_i, S) - \log p(X|S)$$

$$\log p(X|S) = \mathbb{E}_{p(Z_i|X,S)} \log p(X|Z_i, S) - D(p(Z_i|X, S) \parallel p(Z_i|S_i))$$

**Corollary 1**
The sum of the (factual) posterior likelihood and the counterfactual posterior likelihood is lower bounded by

$$\log p(X|S) + \sum_{i=1}^{n} \log p(X'^i|X, S, S') \geq \sum_{i=1}^{n} \log p(X'^n|X, S', S) \\ + \mathbb{E}_{p(Z_i|X,S)} \log p(X|Z_i, S) \\ - D(p(Z_i|X, S) \parallel p(Z_i|S_i)) \\ - \sum_{i=1}^{n} CE(classifier(S_i|Z_i), S_i) \\ + \sum_{i=1}^{n} CE(classifier(S_i|Z_{-i}), S_i) \\ := -\mathcal{L} \tag{3}$$

where $S = S_{-i} \cup S_i$, and $X'^k$ denotes a counterfactual with $k$ many interventions.

**Proof**

The cross entropy of a any classifier of a categorical covariate $S_k$ with finite sample space of size $I$, is non-negative quantity with an upper bound determined by $I$,

$$0 \le CE(\text{classifier}(S_i), S_i) \le f(I) \tag{4}$$

Therefore from Proposition 1 and 4,

$$
\begin{aligned}
\log p(X|S) &= \mathbb{E}_{p(Z_i|X,S)} \log p(X|Z_i, S) - D(p(Z_i|X,S) \parallel p(Z_i|S_i)) \\
&\ge \mathbb{E}_{p(Z_i|X,S)} \log p(X|Z_i, S) - D(p(Z_i|X,S) \parallel p(Z_i|S_i)) \\
&\quad - \sum_{i=1}^{n} CE(classifier(S_i|Z_i), S_i) \\
&\quad - \left( f(I) - \sum_{i=1}^{n} CE(classifier(S_i|Z_{-i}), S_i) \right)
\end{aligned}
\tag{5}
$$

Now adding the term counterfactual likelihood for the n-intervention outcome $\sum_{i=1}^{n} \log p(X'|X, S, S')$ on both hand sides we obtain

$$\log p(X|S) + \sum_{i=1}^{n} \log \hat{p}(X'^{n}|X, S', S) \ge -\mathcal{L} - f(I). \tag{6}$$

Although the cross entropy depends on the classifier and its parameters its upper bound does not. It only depends on the number of possible values of the categorical covariate, and hence it is a constant that can be neglected in optimizing the parameter of our algorithm. This shows that minimizing the objective function in the main text $\mathcal{L}$ will increase the lower bound on the sum of the two log-likelihoods.

**Intuition** After rigorously deriving a loss function as the negative of a lower bound of the sum of log-likelihoods, we provide here intuition for its different terms: The first term indicates the predictability of the original gene expression. Latent variable $Z_i$ is used along with attributes $S_{-i} = S - \{S_i\}$ to predict $X$. The second term indicates the predictability of the counterfactual gene expression. Latent variable $Z_i'$ is used along with attributes $S_{-i} = S - \{S_i\}$ to predict $X_i'$. As shown in Wu et al. (2023), when working with high-dimensional data such as gene expression, it is essential to condition on the original gene expression $X$ and use its individuality to have a high-quality reconstruction of its counterfactual pair $X'$. Accordingly, we use a hierarchy of n encoders and decoders that takes the original gene expression $X$ into account along with given counterfactual attributes to predict $X'$. A causal theoretical motivation behind this term is also explained in the Appendix. The Kullback–Leibler divergence (KL) term serves as a regularization term (Kingma & Welling (2014)). The KL term measures the divergence between the posterior distribution $q(Z_i|X, S)$ and the prior distribution $q(Z_i|S_i)$, conditioning on $S_i$. In our model, $Z_i$ distinguishes between different values of $S_i$, and thus, it conditions on $S_i$ in both the prior and posterior distribution in KL, but not on the effect of other attributes $S_{-i}$. Therefore, the KL term becomes $D_{\text{KL}} q(Z_i|X, S) q(Z_i|S_i)$. The term $\sum_{i=1}^{n} CE(classifier(S_i|Z_i), S_i)$ is the sum of cross entropy losses of the classifiers, indicating that $Z_i$ should contain information about $S_i$. We train a neural network that takes $Z_i$ as input and predicts one-hot $S_i$. For $Z_0$, there is no such term. The last term $\sum_{i=1}^{n} CE(classifier(S_i|Z_{-i}), S_i)$ is the sum of cross entropy losses of the classifiers, indicating that $Z_{-i}$ should have no information about $S_i$. This term serves as the disentanglement of all latent vectors. If the adversarial classifier fails to predict $S_i$, the disentangled representation is achieved. Disentanglement is facilitated in our model by using separate encoders and decoders for each $Z_i$, and when other attributes $S_{-i}$ are fed into decoder $i$ along with $Z_i$, $Z_i$ forgets all of them.

## B    APPENDIX B: HYPERPARAMETER SETTINGS

We use the default hyperparameters for dis2p as same as the VAE module in scvi-tools. Encoders and Decoders all have one 128-dimensional hidden layer, and all latent vectors are 10-dimensional by default. The batch size is set to 128, and the dropout rate is 0.1. We use batch normalization for both decoders and the encoders, but no layer normalization.

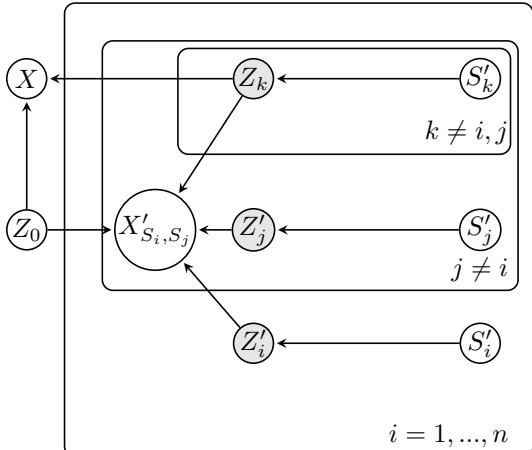

Figure 7: Many worlds graph for counterfactual prediction. There are $n$ worlds each of which is associated with a double intervention. We also explicitly introduce a background hidden variable $Z_0$ which is remains the same across worlds denoting essentially the identity of the individual.

All single-cell methods in the benchmarks are run using their default hyperparameters, but we also did some optimization on their architecture hyperparameters on CPA and biolrd for a fair comparison.

FFVAE and $\beta$-TCVAE are run by default parameters to dis2p and we implemented them similarly to our own method.

