# OpenReview forum: "Disentangling Covariates to Predict Counterfactuals for single-cell data"
_ICLR.cc/2024/Conference — ICLR 2024 Conference Withdrawn Submission_

### Official Review · Reviewer_XHPa · 2023-10-26

**Soundness:** 1 poor
**Presentation:** 1 poor
**Contribution:** 1 poor
**Rating:** 3
**Confidence:** 4

**Summary:**

Here the author's propose _dis2p_, a latent variable model for single-cell sequencing data that learns separate latent spaces for provided covariates as well as a non-covariate-related latent representation. The authors claim that their method provides superior disentanglement performance compared to previously proposed methods for similar problems, and they also claim that this disentanglement can lead to better performance at out-of-distribution prediction tasks.

**Strengths:**

* The authors tackle an important problem of interest in the single-cell community.
* The authors' proposed method is, to my knowledge, novel (though I had significant difficulties understanding the details of the method)
* The authors' proposed method seems more flexible than previously proposed methods for similar problems (e.g. GEARS)

**Weaknesses:**

Unfortunately, due to numerous structural/writing issues with the manuscript, I cannot recommend its acceptance. For example, figure captions are missing/incorrect, notation is inconsistent across the manuscript, and it is often unclear what exactly the authors' experiments are measuring. Thus, after numerous readings I still cannot understand the details of the authors' proposed method and/or the significance of their metrics. Specific issues below:

* **Unclear description of the model:** The authors' proposed framework is quite complicated in terms of network architecture. This is not a problem in and of itself; however, notation for the proposed model is inconsistent across the manuscript, and as such I am still unclear as to how the model operates. For example, the first sentence of Section 2.1 says "We use $n + 1$ encoders parameterized by $\phi_0, \phi_1, \ldots, \phi_n$, yet $\theta_i$ is used to denote the encoder parameters in Figure 2. Moreover, in section 2.2 $\theta_i$ is used to denote _decoder_ parameters, in contradiction with Figure 2. Furthermore, $\psi_i$ is said in Section 2.1 to represent the parameters of another set of encoders, yet is not present at all in Figure 2. Similarly, it's stated that "The input of the encoder $\psi_k$ is $S_k$", yet there are no networks matching this description in Figure 2. Section 2 is thus in need of a thorough rewrite before the manuscript can be published.
* **Duplicated/missing figure captions:** The captions for Figure 1 and Figure 2 are identical, with Figure 2 appearing to be the correct figure for this duplicated caption. This issue further hindered my understanding of the authors' proposed framework.
* **Unclear significance of experimental results:** In Figure 3, the authors perform quantitative benchmarking of their method versus baselines using the SCIB suite of metrics. However, there's no explicit description of what is being measured here. For example: what labels are used to assess "Bio conservation" (cell type? interferon stimulation?) versus "Batch"? Moreover, what latent space are these metrics computed on? The "primary" covariate-invariant latent space or one of the covariate latent spaces? The authors mention that the SCIB metrics are meant to demonstrate disentanglement between the primary/covariate latent spaces, but it's unclear to me how exactly these SCIB results relate to the stated goal of disentanglement.
* **How are hyperparameters tuned?** The authors method introduces three crucial hyperparameters ($\alpha_1, \alpha_2, \alpha_3$) controlling the strength of the different terms in the loss function of Equation (1). However, little to no guidance is given as to how these parameters should be tuned/how values were selected for the presented experimental results, and the results that are provided in Figure 6 are difficult to interpret.  For example, what does each dot represent in the scatter plots of Figure 6 (e.g. different random initializations of the model or different settings of the given hyperparameter)? As-is the results seem extremely noisy, and to my eye it appears that there's near-zero correlation in the "classifier coefficient" plot. This is another example of the lack of clarity in the experimental results mentioned above.


Other minor points that need to be addressed for a final submission:
* Please use parentheses for citations unless the citation is meant to flow with the text (e.g. in the first paragraph "... high-resolution understanding of disease Sikkema et al. (2023a)" --> "... high-resolution understanding of disease (Sikkema et al. (2023a))"
* A citation is missing in the second paragraph in page 4 ("[cite Causality Pearl]")
* Numbers in the "aggregate score" column in Figure 3 are unreadable when the color of the bar gets darker.
* At the top of page 5, the reader is referred to Appendix Note X, which does not exist (I assume this should be Appendix A?)
* What do the authors mean by the equation $R^2 = X$ at the end of the first paragraph in Section 4.3?

**Questions:**

See "Weaknesses" above. As-is I'm very unclear about the inner workings of the authors' proposed model and how to interpret the results of the presented experiments.

---

### Official Review · Reviewer_dmua · 2023-11-01

**Soundness:** 2 fair
**Presentation:** 1 poor
**Contribution:** 2 fair
**Rating:** 3
**Confidence:** 4

**Summary:**

The paper introduces a representation learning model called disentangling covariates to predict counterfactuals (dis2p) for the purpose of decomposing covariate factors in single-cell datasets. Given the complex nature of single-cell analysis and the presence of multiple sources of variability, Dis2P is suggested to separate biological variabilities from undesirable covariates like technical noise. The model comprises a set of *$n$* VAE units, with each unit responsible for learning a latent factor $z_i$, for each covariate $s_i$. The authors have incorporated a group of covariate classifiers to ensure a causal relationship between $z_i$ and $s_i$.

To assess the performance of dis2p, the authors conducted experiments on three scRNA-seq datasets including Heart, Blood, and Liver datasets. They reported disentanglement scores and counterfactual inferences results achieved by dis2p, comparing its performance to other benchmark methods in the single-cell domain, such as scVI and scANVI.

**Strengths:**

The author tackled a challenging and impactful problem in the field of single-cell studies, namely, disentangling the latent space in response to the numerous covariates that induce alterations in the measurement.

**Weaknesses:**

**Major weaknesses:**
- *Novelty:* The paper lacks a clear machine learning contribution, as it doesn't appear to introduce novel ML techniques.

- *Contribution:* The primary contribution of the paper is the learning of a set of latent factors, denoted as $z$, which are influenced by certain covariates. However, this contribution isn't well-supported by the results, and the paper doesn't convincingly demonstrate the superior performance of the proposed model when compared to alternatives (Figure 3).

- *dis2p:*  Despite the fact that the computational cost of training a VAE model for each covariate factor is notably high and inefficient,
 it is not clear why there is need to learn a specific $z$ for each covariate. There's a need to address this issue or provide more convincing arguments for its necessity.

- *Writing and Correctness:* The paper contains multiple typos and errors. The figures' captions lack clarity, and many equations lack proper numbering and explanation.

**Minor weaknesses:**
- In Section 2, the third line references $s^j_i$, and it seems that $i$ should be replaced by $1$.

- In the caption for Figure 1, the variable $z_0$ is not defined. Additionally, the notation $x^{''}_G $should be changed to "$x^{'}_G$."

- The paper should explain how $z_0$ is inferred, as this process is not clear.

- On page 4, in the second paragraph, a citation is missing.

- In the caption for Figure 2, "$A^i_{\zeta}$" and colored distributions are not defined/explained, which can confuse readers.

- The paper presents propositions and corollaries, but these seem to be primarily derivations following variational inference methods.

-- The *covariate classifiers* are essentially regression models predicting $s_i$. Using the term "regression" instead of "classifier" might be more precise.

- The notation for cross entropy, denoted by $CE(.)$ could be improved by using $H(\hat{s}_i, s_i)$.

- In Section 3.2, the text mentions $R2$ values reported for both all genes and the top 20 DEGs, but in Figure 5, it states top 100 DEGs. This discrepancy should be corrected for accuracy and consistency.

**Questions:**

- To elucidate the need for learning $n$ latent factors (for each $s_i$ separately), the authors provided an example involving $z_age$: *"age is just a number but it can manifest in different ways in the skin, brain, heart, etc., all of which are summarized into a latent age variable Z_age."* Are you suggesting that different measurements from the skin, brain, or heart should be mapped to different age values? Or should they all be mapped to the same number? It is not clear why a seperate latent factor is needed for each covariate. Wouldn't it make more sense to combine all covariates and allow the model to learn how to disentangle them?

- On page 4, second paragraph, is it $S^j_i = s_i$ or $S^j_i = S_i$?

- In Section 2.2, what did the authors mean by *" $f$ is a function mapping the latent space to the simplex of the gene expression space."* Do the authors mean $f$ is the probability of success (not being a zero-expressed)? I think the ZINB definition and formulation need to be improved.

- The text mentioned, "*further motivates the need for the latent variable $Z_0$ that summarizes the identity and individuality of the cell."* It is not clear what is meant by "identity" here. Did the authors mean the cell type identity?

- In Figure 2a, why does the decoder, $D_{\phi_0}$, receive the covariate vector $S$? Shouldn't it be trained to be independent of covariates?

- In Section 2.4, how is the equality derived?

- In Eq 1, does $D$ denote KL divergence? It might be better to use a different notation for distance, as the decoders are also denoted by $D$.

- In Figure 6, what do the colors represent? What are "counterfactual" and "classifier coefficient"?

- How do the reported values in Figure 3 demonstrate the superior performance of dis2p?

---

### Official Review · Reviewer_bDut · 2023-11-01

**Soundness:** 2 fair
**Presentation:** 1 poor
**Contribution:** 2 fair
**Rating:** 3
**Confidence:** 4

**Summary:**

This paper presented a deep learning framework for counterfactual generation while learning disentangled representations for observed factors and is presented in the context of single-cell data where the observed factors are perturbations and cell covariates. The objective optimizes outcome likelihood and representation for observed factors together in an end-to-end fashion. Disentanglement between representations is achieved by enforcing separate prior, and incorporating classfication loss and adversarial loss. Experiments show better results on disentanglement metrics and counterfactual generation compared to some benchmarks.

**Strengths:**

Literature review was done thoroughly. Visualization of experiments are presented clearly and easy to follow.

**Weaknesses:**

The paper is pretty hard to follow aside from the experiment results. The language used and definitions seem very informal. Some notations are used before defined (e.g. I have to read [1] to understand what ' as a superscript of a variable means, $g$ in the equations in cell encoder section is not defined and I can't tell which ones are parameterized with $\phi$ and $\theta$). Some reference are not cleaned up before submission (e.g. "[cite Causality Pearl]"). Firgure 1 and 2 are very raw and clearly hasn't been polished. It's hard to understand the workflow from Figure 2. Overall, the manuscript seem unready.

As of the proposed method, I think evaluating the disentangling between $Z_i$ ($i > 0$) instead of $Z_i$ ($i \geq 0$) is meaningless. Disentangling the representations of observed factor $S_i$ ($i > 0$) is not a hard problem to be dealt with, obviously you can just enforce them on separate conditional prior as you did and they are naturally disentangled by design. Prior work like [2] have already done this. I can propose a model where the latent is $(\mathbf{Z}_0, Z_1, Z_2, ... Z_n)$ where I simply have $Z_1 = S_1$, $Z_2 = S_2$, ..., $Z_n = S_n$, then my model is going to beat yours and all your benchmarks on your MI metrics. The problem with yours and [2] and what I just proposed is that $\mathbf{Z}_0$, the true exogenous noise, is not disentangled from $Z_i$ ($i > 0$). It's still free to encode any information related to the observed factors. To use your example at the end of page 3, all of those factors like skin, brain, heart that are manifested by age, can still be encoded in $\mathbf{Z}_0$. In [1], they have a divergence term that prevents this problem from happening. I do not see any loss term or architecture design in your framework to ensure or encourage that.


[1] Wu, Yulun, et al. "Variational causal inference." arXiv preprint arXiv:2209.05935 (2022).

[2] Shen, Xinwei, et al. "Weakly supervised disentangled generative causal representation learning." The Journal of Machine Learning Research 23.1 (2022): 10994-11048.

**Questions:**

How is $\hat{p}(X' | X, S', S)$ evaluated if $X'$ is not observed?

The manuscript feels rushed and I hope the authors have taken this time between submission and rebuttal to polish the paper and submit a better revised version. I'm open to raise the score after improvements are shown.

---

### Official Review · Reviewer_Woon · 2023-11-10

**Soundness:** 1 poor
**Presentation:** 1 poor
**Contribution:** 2 fair
**Rating:** 3
**Confidence:** 4

**Summary:**

The authors introduce an autoencoder-structure to infer structured latent spaces for single cell data with the aim to be able to perform "counterfactual inference".
The intent of the authors is to propose a model which given n discrete covariates learns latent spaces Z_n and explains away the effect of said covariate. They propose an encoder-decoder structure which blends some inspiration from variational inference and some classifiers together, and suggest that their method performs this inference.

They demonstrate their technique on three scRNA seq datasets, heart, blood, and liver.

**Strengths:**

The key idea is interesting, in that one may want a model to be able to simulate outcomes over cells given covariates.
However, this purely depends on specifying a well-posed generative model depending on the covariates, which is not a hard task in itself.

Another strength of the paper is that the authors identify an interesting scenario where counterfactual cells may be desireable under different observed properties like age and time etc. to capture invariant properties of cells.

**Weaknesses:**

Unfortunately the paper exudes messiness.

Starting from the model specification and the figures.
For example, the legend of Figure 1 does not describe what is in it appropriately, neither are the two subfigures even labeled a and b,.

And to the heart of the problem: the model and inference here and insufficiently specified and executed.
First, the authros start by specifying encoders for a model they have not introduced yet.
The language throughout section 2.1 and 2.2 is very misleading, iterating between claims of "encoders predicting vectors X" to using a likelihood function which clearly contradicts the idea that vectors X are directly predicted.
In 2.1 the authors claim to be "using encoders to be inferring prior distributions q_psi_k (Z_k | S_k)".
These appear not to prior distributions, they are approximate variational families. What is the prior here? And priors are commonly not inferred, but specified, unless we explicitly assume Type II maximum likelihood or some such inferential procedure which is not mentioned here. I must say I could not get to the bottom of what is really being computed here.

Another complication is that the authors claim that they generate observations X for each of the n decoders in Sec. 2.2.
How are these brought together? Let alone how do they blend in the likelihood?

In Sec. 2.3 the authors discuss using classifiers to encourage disentanglement and add them to their loss.
This feels unprincipled and not well explained or justified. I read the math carefully, and it breaks the idea of specifying a clear forward model that can be inverted here, by seemingly attaching unjustified losses together and then claiming these achieve disentanglement and causality. I have no doubt the intention is to create causally invariant subspaces, but when aiming for such a goal the writing and mathematical precision need to have a higher bar.

In Sec 2.4 the authors use the model p(X' | Z, S-k) which ahs never been introduced. The only model specification exists via Figure 1 which has a model with a factor p(X|Z), so it is hard to tell what the object in the Equation in Sec. 2.4 is doing.
I also want to add that it appears to not be correctly specifying a counterfactual, since Z is inferred using the control, and then reused for the case by just concatenating S_k. I again have no doubt the intention is to write down a distribution which IS the counterfactual, but this does not read like that exactly.


Overall, the writing here is really unclear and not up to par for an ICLR paper, and the authors use various terms inappropriately when paired with the math the use (i.e. prior, causality, etc.).

The proicess described on page 6 in Sec.2.4 is also messy, and I would strongly suggest adding two algorithms boxes: one showing what should happen, and one showing what the authors do during training.

Finally, the loss function suffers from the blend of classifiers and probabilistic models without a clear modeling idea connecting them.
While it would be ok to do this if one were to be modeling images wanting to maximize perceptual similarity, the claim of achieving causality here makes the interpretation of the used method in terms of a clear forward process and its inversion paramount.

Finally, I must say the experiments are not convincing. The baseliens are poorly chosen, for instance a simple baseline of a conditional VAE which would be a trivially invertible generative model permitting counterfactual reasoning at great ease is not included, the other models used are idiosynchratic models, and the metric used are surprising. Where is some metric showing the basic generative capability of the models? How is the likelihood being consumed to generate these metrics? How do these metrics properly link to the predictive distribution during testing?

The mathematical exposition of the paper could benefit greatly from improved notation, dimensions, and overall consistentcy.
I recommend the authors carefully look over their used indices. For instance, n indicated the number of covariates, but N is the number of samples in the dataset. All of this put together with the fact that sometimes indices are not even used and it is unclear whether all of Z or just one part of it is being changed leads to major mathematical confusion giving a messy impression.

One key modeling concern beyond the exposition and quality of the technical pieces that I have is as follows:
The authors assume that each covariate S_n leads to an independent latent variable Z_n explaining away everything that this covariate causes. It feels like a very strong independence assumption, when it is probably more reasonable to consider that various aspects of Z could share information with various covariates. For instance, in a real experiment consider two covariates: plate id and age.
If in a large enough experiment the plates are processed sequentially and the procedure to proccess them takes long, plate ID and age are partially colliding by being proxies to each other by virtue of experimental bias (i.e. the first plate is the oldest, so all cells on that plate are old, so age is a proxy for plate ID since the experimental design is sequential in nature). Forcing independent latent variables for age and plate ID implies that the counterfactual for those would not really be able to disentangle these effects, since the model fails to account for that correlation. I thus suspect that even if this model had been executed well here, it would not be a particularly good model of typical effects seen in such systems.
As such I also do not find the model setup by itself sufficiently close to a realistic case in cell biology for its stated purpose.


Minor nitpick:
A lot of typos or citation errors abound in the paper.
An example is on page 4 at the top there is a placeholder "[cite Causality Pearl]".
Another nitpick is also that Figure 3 is not really legible and needs to be simplified.

I consider these minor issues, but recommend the authors do a proper cleanup.

**Questions:**

While I agree with the goals of the paper, it  needs a lot of work.
The math is not always clear and consistent, the writing is not polished enough to be internally consistent within the goals of the paper and also not consistent with the typical meaning of some of the tools used.

My ask to the authors would be to really carefully clarify what their actual forward model is, and what the inference ends up being and to rewrite the paper to reflect that.
Once that is achieved, one can reason about counterfactual reasoning given that clarified modeling assumption.

For any model that uses a likelihood model and is supposed to utilize a generative interpretation, it is key to assess its quality as a generative model fairly, and the authors do not do that. I recommend introduced numerous baselines to conduct such tests while keeping the likelihood consistent.